# Non-Invasive Assessment Method Using Thoracic-Abdominal Profile Image Acquisition and Mathematical Modeling with Bezier Curves

**DOI:** 10.3390/jcm8010065

**Published:** 2019-01-09

**Authors:** Monica Ana Paraschiva Purcaru, Angela Repanovici, Tiberiu Nedeloiu

**Affiliations:** 1Department of Mathematics and Informatics, Faculty of Mathematics and Informatics, Transilvania University of Brasov, 500036 Brasov, Romania; mpurcaru@unitbv.ro; 2Department of Product Design, Mechatronics and Environment, Faculty of Product Design and Environment, Transilvania University of Brasov, 500036 Brasov, Romania; 3Department of Medical and Surgical Specialization, Faculty of Medicine, Transilvania University of Brasov, 500036 Brasov, Romania; nedeloiu.t@unitbv.ro

**Keywords:** Bézier curves, curvature, arc length, area, non-alcoholic fatty liver disease, ultrasound, waist circumference, anthropometry, hepatomegaly, non-invasive methods

## Abstract

The study was performed at Brasov County Hospital, in the Internal Medicine, Diabetes, Gastroenterology and Cardiology Wards with the collaboration of Transylvania University of Brasov, as a study approved by the Ethical Board of the university. The study aimed at assessing the connection between the anthropometric parameters of abdominal adiposity (measured by means of an original experiment designed to determine the curvature of the thoracic-abdominal adiposity for the patients and processed by help of a mathematical model based upon Bezier curves geometry) and the fat load of the liver (assessed by ultrasound by measuring the diameters of both hepatic lobes) for the patients diagnosed with Non-Alcoholic Fatty Liver Disease (NAFLD). The existence of ten types of thoracic-abdominal curves profiles were statistically analyzed in order to evaluate in a simple manner the liver size in NAFLD. The method of diagnosis is based on an easily reproduced experiment, it is original, innovative, non-invasive, and cost-effective. Can be implemented anywhere in the world, there is no need for investment, only for determining the profile of the belly.

## 1. Introduction

### 1.1. Hepatic Steatosis: Investigation Methods

Hepatic steatosis is the build-up of lipids within hepatocytes. It is the simplest stage in non-alcoholic fatty liver disease (NAFLD). It occurs in approximately 30% of the general population and as much as 90% of the obese population in the United States [1]. It may progress to non-alcoholic steatohepatitis, which is a state of hepatocellular inflammation and damage in response to the accumulated fat [2]. Liver biopsy remains the gold standard tool to diagnose and stage NAFLD [3], but is an invasive method with the risk of complications ranging from simple pain to life threatening bleeding. and is also associated with sampling error [1].

#### 1.1.1. Non-Invasive Methods

Non-invasive assessment of steatosis and fibrosis is of growing relevance in non-alcoholic fatty liver disease (NAFLD), but most of the imaging investigations are expensive and laborious. 1H-Magnetic resonance spectroscopy (1H-MRS) and the ultrasound-based controlled attenuation parameter (CAP) correlate with biopsy proven steatosis, but have not been correlated with each other so far [4]. In Karlas et al. study a head-to-head comparison performed between both methods is presented. In conclusion data suggest a comparable diagnostic value of CAP and 1H-MRS for hepatic steatosis quantification. Combined with the simultaneous transient elastography (TE) for fibrosis assessment, CAP represents an efficient method for non-invasive characterization of NAFLD. Limited correlation between CAP and 1H-MRS may be explained by different technical aspects, anthropometry, and presence of advanced liver fibrosis [4].

Other radiological investigations for non-invasive diagnosis of NAFLD have been proposed, including computed tomography, magnetic resonance imaging and proton magnetic resonance spectroscopy [5], the controlled attenuation parameter using transient elastography [6] and Xenon 133 scan [7].

Ultrasound is a non-invasive, widely available, and accurate tool in the detection of NAFLD [8] when sonographic features unique to NAFLD are standardized [9] and used to aid in diagnosis.

#### 1.1.2. Measurement of Body Fat and Fat Distribution

Studies have shown that obesity is strongly associated with hepatic steatosis [10]. Therefore, the usual management of NAFLD includes gradual weight reduction and increase in physical activity [11]. However, it remains uncertain whether excessive food consumption per se causes fatty liver or diets that are enriched in certain types of food are more likely to cause hepatic steatosis [12,13]. Body mass index (BMI) has been directly linked with the prevalence of NAFLD [13]. This leads to the speculation that a greater BMI in patients with NAFLD will lead to a more severe degree of hepatic steatosis [14]. NAFLD now occurs in the range of 65–92.3% of morbidly obese patients (BMI > 40 kg/m^2^) [15].

#### 1.1.3. Waist Circumference (WC) Is a Simple and Inexpensive Tool for Assessing Body Fat Distribution

WC correlates well with abdominal obesity and it is associated with increased risk for adiposity-related morbidity and mortality [4]. WC and waist/hip ratio (WHR) are used as markers of abdominal obesity as they reflect central obesity [15]. It has been proposed that WHR and/or WC are more related to NAFLD than BMI [16].

In summary, WHR and WC are simple tools that can be applied as important anthropometric indicators to screen populations with a high risk for NAFLD [1].

### 1.2. Bezier Curves

#### 1.2.1. Theoretical Consideration

Bézier curves were introduced due to practical requirements that occurred in engineering in 1959 by the French mathematician and physicist Paul de Casteljau. While working at Citroën, he introduced an algorithm of evaluating calculus for a certain family of curves. After a few years, the French engineer Paul Bézier investigated independently the same class of curves, providing in 1962 the parametrical equations of the curves bearing his name.

Bézier curves have remarkable geometric properties and are used for modeling continuous and derivable curves. Some examples of using Bézier curves are presented in the following. A cubic Bézier curve may be used to describe the profiles of horns for high displacement amplification and ease of machining [17]. Bézier curves, bioexponential and cubic fitted to the Opto-Electronic Conversion Function curves reconstructed for the tested cameras [18]. The fitting procedure is described by Murthy et al. [19] and provides a set of smooth Bézier curves that describe the isopotentials. An alternative approach to curvature estimation involves precomputing the generic form of the curvature function for a Bézier curve [19,20].

#### 1.2.2. Bezier Curves, Mathematical Formulation

As in [20,21,22], let us consider $P_{0},P_{1},\ldots ,P_{n}$, *n* + 1 distinct points in the Euclidean two-dimensional space *E*^2^, named control points or just controls. The polygon obtained by joining the control points starting with *P*_0_ and finishing with *P_n_* is called control polygon or Bézier polygon. The control polygon is not unique.

**Definition** **1.**
*Bézier curve of n grade corresponding to controls*
$P_{0},P_{1},\ldots ,P_{n}$
*is:*
(1)$$B\left(t\right)=\overset{n}{\underset{k=0}{\sum }}P_{k}b_{k,n}\left(t\right)$$
*where*
$b_{k,n}(t)$
*are Bernstein polynomials of n degree and are given by:*
(2)$$b_{k,n}\left(t\right)=C_{n}^{k}t^{k}{\left(1-t\right)}^{n-k},t\in \left[0,1\right]$$


**Observation** **1.**
*The spirit of construction of relation (1) has inspired the proof techniques in opinion dynamics in social networks [23].*


**Example** **1.**
*For n = 2, a quadratic Bézier curve is defined by the position of three control points, (P_0_, P_1_, P_2_) defining a quadratic Bézier curve (3) and it is constructed by evaluating an independent parameter t in [0, 1] interval:*
(3)$$B\left(t\right)={\left(1-t\right)}^{2}P_{0}+2t\left(1-t\right)P_{1}+t^{2}P_{2},t\in \left[0,1\right]$$


**Observation** **2.**
$B\left(0\right)=P_{0}$
*and*
$B\left(1\right)=P_{n}$

*Properties of Bézier curves:*
*1*.*The ends of the curve are tangent to the segments*$\left[P_{0}P_{1}\right]$*respectively*$\left[P_{n-1}P_{n}\right]$;*2*.
*A Bézier curve can be divided into two Bézier curves in any of its points;*
*3*.
*Bézier curve is entirely included in the convex cover of its control points;*
*4*.
*A quadratic Bézier curve is a parabola segment;*
*5*.
*By translating or rotating a Bézier curve, another Bézier curve is obtained.*



## 2. Experimental Section

### 2.1. Ethical Statement

The present study was conducted in accordance with the guidelines of Transylvania University Ethical Commission (Approval B21-2013) and under Spitalul Municipal Brasov, Romania [24].

Non-alcoholic fatty liver disease (NAFLD) is considered a form a visceral adiposity. The fat deposits (triglycerides) in the liver are followed by hepatomegaly, proportional to the deposited quantity of fats [25]. There are strong positive correlations between cardiovascular risk, the values of plasmatic lipids, amount of visceral adiposity (fat around intra-abdominal organs) and the amount of subcutaneous adiposity at abdominal level [26,27].

Direct evaluation of the amount of visceral and abdominal adiposity requires consuming and expensive imaging investigations (computed tomography, nuclear magnetic resonance etc.). This is the reason for searching indirect possibilities of measuring them. Currently, the most used parameter correlated with the amount of abdominal adiposity is waist circumference. WC measured according to WHO recommendations [11] in a plane which is parallel to the floor plane, at half the distance between the last rib and the iliac crest, at the end of expiration. The patient with empty stomach (à jeun), stays with close heels and relaxed abdominal muscles. This way to measure WC underestimates the relationship with abdominal obesity, the most part of adiposity being placed under the waist circle, due to different shapes of the “abdominal profile” (Figure 1).

### 2.2. Experimental Method of Designing Thoracic-Abdominal Profile

With recent advances in optical and digital technology, the consumer-level digital camera has become a convenient and cost effective instrument for acquiring images for quantitative analysis. One major issue with using consumer-level cameras is obtaining a linear response, which is a prerequisite for tasks like profile curves.

Thoracic-abdominal profile curve is obtained by acquiring the profile image with a digital camera of 6 megapixels (MP) (Sony Cyber Shot 6.0. MP camera, Sony Corporation, Japan, Tokio), fixed on a height adjustable tripod. Acquiring the thoracic-abdominal profile image of the patients was performed at 1m distance. The optic axis of the camera was focused at waist level for each subject (height of WC measurement), focalization was adjusted according to the patients’ anthropometric characteristics using the adjustable height of the tripod. Thus, the optical parallax errors were minimized and have not been calculated. Standard photos were obtained for each patient. The vector graphics program Adobe Illustrator (using Bézier curves) was used in order to overlap the obtained image and draw a vertical line starting from the anterior superior iliac spine (iliac spine is palpated and dotted with a marker on the patient skin—white “x”), to the anterior axillary fold from where a perfectly overlapped curve with the thoracic-abdominal profile descends and returns to the starting point (Figure 2).

Thus a four point defined curve is described (Figure 3): $P_{0}$ point represented on the anterior fold of the arm, $P_{1}$ the most prominent point of the abdomen (extreme point of the curve ($P_{0}P_{3}$)), $P_{4}$ inferior point of the profile (extreme point of the curve ($P_{3}P_{2}$)) and $P_{2}$ the point where the vertical descending from $P_{0}$ crosses the iliac crest.

Curve ($P_{0}P_{1}P_{4}P_{2}$) is obtained by linking two quadratic Bézier curves: ($P_{0}P_{3}$), with the controls: $P_{0}$, $P_{5}$, $P_{3}$ and ($P_{3}P_{2}$), with the controls: $P_{3}$, $P_{6}$, $P_{2}$.

Characteristics of the Bézier curves created by the used vector graphics program were evaluated based upon differential geometry rules. By help of the Adobe Illustrator program the coordinates of points $P_{0},\;P_{1},\;P_{2},\;P_{3},\;P_{4},\;P_{5},\;P_{6},\;P_{7},\;P_{8}$ were gathered and then used in the following calculus. They were denoted: $P_{0}\left(0,0\right)$, $P_{1}\left(x_{1},y_{1}\right)$, $P_{2}\left(0,y_{2}\right)$, $P_{3}\left(x_{3},y_{3}\right)$, $P_{4}\left(x_{4},y_{4}\right)$, $P_{5}\left(x_{5},y_{5}\right)$, $P_{6}\left(x_{6},y_{6}\right)$, $P_{7}\left(x_{7},y_{7}\right)$, $P_{8}\left(0,y_{1}\right)$.

A database for 100 patients was created, whose fields are filled up for each patient with the characteristics of their own thoracic-abdominal profile curve, determined by help of the formulas: (5), (16). Beside the information provided by this database, in order to process them as efficiently as possible and reach a simplified clinical interpretation, a classification of the thoracic-abdominal profile curves in three categories was achieved considering one or two of the geometric criteria.

The measuring units are conventional and depend upon the resolution of the computer’s display.

**Observation** **3.**
*If the minimum point is at P_2_ instead of P_4_, then Bezier curves method can still be used analogously.*


### 2.3. Research Mathematical Model Based on Bezier Curves

The mathematical model used in the present paper is based on the linked quadratic Bézier parametric curves, named paths in the graphics program, Adobe Illustrator.

Construction of a Bézier curve can be accomplished in two manners: either using geometric methods (Paul de Casteljau algorithm), or using algebraic methods (Bernstein form). The present paper uses the first recursive manner, by means of which starting form three points called control points, we aim at building curves that are as similar as possible to the curves resulted from the practical applications.

A limit of the considered mathematical model is the lack of a fitting procedure which optimizes the intermediate control points to minimize the least squared error, meaning the lack of an algorithm optimization procedure.

In parametric form each coordinate of a point on a curve is represented as a function of a single parameter [19]. Relation (3) is equivalent to the Equations (4), meaning the parametric equations of a Bézier quadratic curve:(4)$$\{\begin{array}{l} x\left(t\right)={\left(1-t\right)}^{2}x_{P_{0}}+2t\left(1-t\right)x_{P_{1}}+t^{2}x_{P_{2}}\\ y\left(t\right)={\left(1-t\right)}^{2}y_{P_{0}}+2t\left(1-t\right)y_{P_{1}}+t^{2}y_{P_{2}} \end{array},t\in \left[0,1\right]$$

In this investigation using computer aided graphics, we aim at building images of the thoracic-abdominal profile curves of some patients, very similar to reality. Each Bézier curve is a second degree polynomial.

#### 2.3.1. Calculus of Thoracic-Abdominal Curve Parameters

Using Equations (4), the parametric equations of Bézier curves ($P_{0}P_{3}$) and ($P_{3}P_{2}$) expressions are given in relations (5) and (6), respectively:(5)$$\left(P_{0}P_{3}\right):\{\begin{array}{l} x\left(t\right)=2t\left(1-t\right)x_{5}+t^{2}x_{3}\\ y\left(t\right)=2t\left(1-t\right)y_{5}+t^{2}y_{3} \end{array},t\in \left[0,1\right],$$
(6)$$\left(P_{3}P_{2}\right):\{\begin{array}{l} x\left(t\right)={\left(1-t\right)}^{2}x_{3}+2t\left(1-t\right)x_{6}\\ y\left(t\right)={\left(1-t\right)}^{2}y_{3}+2t\left(1-t\right)y_{6}+t^{2}y_{2} \end{array},t\in \left[0,1\right],$$

Or equivalently:(5a)$$\left(P_{0}P_{3}\right):\{\begin{array}{l} x\left(t\right)=t^{2}\left(x_{3}-2x_{5}\right)+2x_{5}t\\ y\left(t\right)=t^{2}\left(y_{3}-2y_{5}\right)+2y_{5}t \end{array},t\in \left[0,1\right],$$
(6a)$$\left(P_{3}P_{2}\right):\{\begin{array}{l} x\left(t\right)=t^{2}\left(x_{3}-2x_{6}\right)+2t\left(x_{6}-x_{3}\right)+x_{3}\\ y\left(t\right)=t^{2}\left(y_{2}+y_{3}-2y_{6}\right)+2t\left(y_{6}-y_{3}\right)+y_{3} \end{array},t\in \left[0,1\right].$$

Using Equations (5a) and (6a), respectively and the coordinates of the extreme points: $P_{1}\left(x_{1},y_{1}\right)$ and $P_{4}\left(x_{4},y_{4}\right)$, we determine from Equations (7) and (8), respectively the parameter *t* corresponding to each extreme point: $P_{1}$, for the Bézier curve ($P_{0}P_{3}$) and $P_{4}$, for the Bézier curve ($P_{3}P_{2}$), respectively $t_{1}$ and $t_{4}$:
(7)$$t_{1}=\frac{x_{1}y_{3}-2x_{1}y_{5}+2x_{5}y_{1}-x_{3}y_{1}}{2\left(x_{5}y_{3}-x_{3}y_{5}\right)}$$
(8)$$t_{4}=\frac{x_{4}y_{2}+x_{4}y_{3}-2x_{4}y_{6}-x_{3}y_{2}+2x_{3}y_{6}+2x_{6}y_{4}-2x_{6}y_{3}-x_{3}y_{4}}{2\left(x_{6}y_{2}-x_{6}y_{3}-x_{3}y_{2}+x_{3}y_{6}\right)}$$

Using Equations (5a) and (6a), respectively and the calculus formula of the curvature for a plane curve, in the extreme points: $P_{1}\left(x_{1},y_{1}\right)$ and $P_{4}\left(x_{4},y_{4}\right)$, in the relations (9a) and (10a), respectively, the curvature $K_{1}$, in $P_{1}$, is determined for the Bézier curve ($P_{0}P_{3}$) and curvature $K_{2}$, in $P_{4}$, for the Bézier curve ($P_{3}P_{2}$):
(9)K(t)=2[(x3−2x5)t+x5](y3−2y5)−2(x3−2x5)[(y3−2y5)t+y5]{[(x3−2x5)t+x5]2+[(y3−2y5)t+y5]2}32

From where:(9a)$$K_{1}=K\left(t_{1}\right)=K\left(t\right)|P_{1}$$
(10)K(t)=2[(x3−2x6)t+x6−x3](y2−y3−2y6)−2(x3−2x6)[(y2+y3−2y6)t+y6−y3]{[(x3−2x6)t+x6−x3]2+[(y2+y3−2y6)t+y6−y3]2}32

And then:(10a)$$K_{2}=K\left(t_{4}\right)=K\left(t\right)|P_{4}$$

Using relations (5a) and (6a), respectively and the calculus formula of the arc length for a plane curve, we are able to determine from (11), (11a) and (11b) respectively (12), (12a) and (12b) the calculus formulas of the arc length ($P_{4}P_{2}$), InfArc, respectively the arc length ($P_{0}P_{1}$), SupArc:(11)$$L_{arcP_{4}P_{2}}=\int_{t_{4}}^{1}\sqrt{{\dot{x}}^{2}\left(t\right)+{\dot{y}}^{2}\left(t\right)}dt=2\int_{t_{4}}^{1}\sqrt{at^{2}+bt+c}dt$$
where: (11a)$$\left\{ \begin{array}{l} a=x_{3}^{2}+4x_{6}^{2}-4x_{3}x_{6}+y_{3}^{2}+y_{2}^{2}+4y_{6}^{2}+2y_{2}y_{3}-4y_{3}y_{6}-4y_{2}y_{6},\\ b=2(3x_{3}x_{6}+3y_{3}y_{6}-2x_{6}^{2}-2y_{6}^{2}-x_{3}^{2}-y_{3}^{2}+y_{2}y_{6}-y_{2}y_{3}),\\ c=x_{6}^{2}+x_{3}^{2}-2x_{3}x_{6}-2y_{3}y_{6}. \end{array}\right.$$
(11b)$$InfArc=\sqrt{a}\left(e\sqrt{e^{2}+k^{2}}-d^{\prime }\sqrt{d^{\prime 2}+k^{2}}+k^{2}\ln\left(e+\sqrt{e^{2}+k^{2}}\right)-k^{2}\ln\left(d^{\prime }+\sqrt{d^{\prime 2}+k^{2}}\right)\right)$$
where:$$e=1+\frac{b}{2a},d^{\prime }=t_{4}+\frac{b}{2a},k^{2}=-\frac{\Delta^{\prime }}{a^{2}},\;\Delta^{\prime }=y_{3}^{2}y_{6}^{2}+4y_{6}^{4}+y_{3}^{4}+y_{2}^{2}y_{6}^{2}+y_{2}^{2}y_{3}^{2}+2x_{3}x_{6}y_{3}y_{6}-12x_{3}x_{6}^{3}-4x_{3}x_{6}y_{6}^{2}-4x_{3}x_{6}y_{3}^{2}\;-2x_{3}x_{6}y_{2}y_{6}-x_{3}x_{6}y_{2}y_{3}-4y_{3}y_{6}^{3}-4y_{3}^{3}y_{6}+2y_{2}y_{3}y_{6}^{2}-4y_{2}y_{3}^{2}y_{6}+4x_{6}^{2}y_{6}^{2}+3x_{6}^{2}y_{3}^{2}\;+2x_{6}^{2}y_{2}y_{3}-4y_{2}y_{6}^{3}+x_{3}^{2}y_{3}^{2}+2x_{3}^{2}y_{2}y_{6}+2y_{2}y_{3}^{3}+12x_{3}x_{6}^{3}-x_{6}^{2}y_{2}^{2}-x_{3}^{2}y_{2}^{2}+2x_{3}x_{6}y_{2}^{2}.$$
(12)$$L_{arcP_{0}P_{1}}=\int_{0}^{t_{1}}\sqrt{{\dot{x}}^{2}\left(t\right)+{\dot{y}}^{2}\left(t\right)}dt=2\int_{0}^{t_{1}}\sqrt{at^{2}+bt+c}dt$$
where:(12a)$$\{\begin{array}{l} a=x_{3}^{2}-4x_{3}x_{5}+4x_{5}^{2}+y_{3}^{2}-4y_{3}y_{5}+4y_{5}^{2},\\ b=2\left(3x_{3}x_{5}+y_{3}y_{5}-2x_{5}^{2}-2y_{5}^{2}\right),\\ c=x_{5}^{2}+y_{5}^{2}. \end{array}$$
(12b)$$SupArc=\sqrt{a}\left(d\sqrt{d^{2}+k^{2}}-e\sqrt{e^{2}+k^{2}}+k^{2}\ln\left(d+\sqrt{d^{2}+k^{2}}\right)-k^{2}\ln\left(e+\sqrt{e^{2}+k^{2}}\right)\right)$$
where:$$d=t_{1}+\frac{b}{2a},e=\frac{b}{2a},k^{2}=-\frac{\Delta }{4a},\;\Delta =-3x_{3}^{2}x_{5}^{2}-3y_{3}^{2}y_{5}^{2}-12x_{5}^{4}-12y_{5}^{4}+2x_{3}x_{5}y_{3}y_{5}+12x_{3}x_{5}^{3}+12x_{3}x_{5}y_{3}y_{5}^{2}+12x_{5}^{2}y_{3}y_{5}+12y_{3}y_{5}^{3}-24x_{5}^{2}y_{5}^{2}-4x_{5}^{2}y_{3}^{2}-4x_{3}^{2}y_{5}^{2}.$$

From (5a) and (6a), respectively and the calculus formula of the area below a function graphic, we determine from (13) and (15), respectively the calculus formulas for the areas limited by the curve ($P_{0}P_{1}$) and the vertical line $P_{0}P_{2}$: (Area1), and by curve ($P_{1}P_{2}$) and vertical line $P_{0}P_{2}$: (Area2), respectively (Figure 3):(13)$$Area1=\frac{ay_{1}^{3}}{3}+\frac{by_{1}^{2}}{2}$$
where:(14)$$\{\begin{array}{l} a=\frac{x_{3}y_{1}-x_{1}y_{3}}{y_{1}y_{3}^{2}-y_{1}^{2}y_{3}},\\ b=\frac{x_{3}}{y_{3}}-ay_{3}. \end{array}$$
(15)$$Area2=S_{total}+Area1$$

The plane separating Area1 from Area2 is the plane of the waist circumference (CiTal).

Using relations (5a), (6a) and formula of the rotation body volume determined by the arc $P_{1}P_{2}$ around the axis $P_{0}P_{2}$, we determine from (16) the volume of the “lower belly”.
(16)$$V_{low}=\frac{\pi }{2}\left(\frac{2a^{2}}{5}y_{4}^{5}+aby_{4}^{4}+\frac{2b^{2}}{3}y_{4}^{3}-\frac{a^{2}}{5}y_{1}^{5}-\frac{ab}{2}y_{1}^{4}-\frac{b^{2}}{3}y_{1}^{3}-\frac{a^{2}}{5}y_{2}^{5}-\frac{ab}{2}y_{2}^{4}-\frac{b^{2}}{3}y_{2}^{3}\right)$$
where *a*, *b* are constants given in (14).

Using Equations (5a) and (6a), respectively and the calculus formula of the osculating circle radius for a plane curve, in the extreme points: $P_{1}\left(x_{1},y_{1}\right)$ and $P_{4}\left(x_{4},y_{4}\right)$, we are able to determine from (17) and (18), respectively the radius of the osculating circle $R_{1}$, in $P_{1}$, for the Bézier curve ($P_{0}P_{3}$) and the radius of the osculating circle $R_{2}$, in $P_{4}$, for the Bézier curve ($P_{3}P_{2}$):
(17)$$R_{1}=\left(\frac{1}{\frac{\left|2\left[\left(x_{3}-2x_{5}\right)t+x_{5}\right]\left(y_{3}-2y_{5}\right)-2\left(x_{3}-2x_{5}\right)\left[\left(y_{3}-2y_{5}\right)t+y_{5}\right]\right|}{{\left\{{\left[\left(x_{3}-2x_{5}\right)t+x_{5}\right]}^{2}+{\left[\left(y_{3}-2y_{5}\right)t+y_{5}\right]}^{2}\right\}}^{3/2}}}\right){|}_{P_{1}}$$
(18)$$R_{2}=\left(\frac{1}{\frac{\left|2\left[\left(x_{3}-2x_{6}\right)t+x_{6}-x_{3}\right]\left(y_{2}-y_{3}-2y_{6}\right)-2\left(x_{3}-2x_{6}\right)\left[\left(y_{2}+y_{3}-2y_{6}\right)t+y_{6}-y_{3}\right]\right|}{{\left\{{\left[\left(x_{3}-2x_{6}\right)t+x_{6}-x_{3}\right]}^{2}+{\left[\left(y_{2}+y_{3}-2y_{6}\right)t+y_{6}-y_{3}\right]}^{2}\right\}}^{3/2}}}\right){|}_{P_{4}}$$
where $R_{i}=\frac{1}{K_{i}},$
*i* = 1, 2.

#### 2.3.2. Classification of Abdominal Curves

One first classification has been made according to the curvature $K_{2}$ in $P_{4}$ and the arc length ($P_{4}P_{2}$), InfArc.
$$I_{1}\{\begin{array}{l} K_{2}\in \left[0.000493763;\;0.027066286\right]\\ InfArc\in \left[108.0939;214.7204\right] \end{array}\;\mathrm{raised}\;\mathrm{profile}\;(\mathrm{RP})$$

For the thoracic-abdominal profile curve the arc (*P*_3_*P*_2_) is almost identical to the segment [*P*_3_*P*_2_], the belly is straight on the lower side, meaning the belly is very little fallen;
$$I_{2}\{\begin{array}{l} K_{2}\in \left[0.029424926;\;0.080514181\right]\\ InfArc\in \left[225.8813;391.5648\right] \end{array}\;\mathrm{fallen}\;\mathrm{profile}\;(\mathrm{FP})$$

The thoracic-abdominal profile curve follows the arcs (*P*_3_*P*_4_), respectively (*P*_4_*P*_2_) very close in the vicinity of point *P*_4_ meaning the belly is very “pursed” at the bottom.
IThe second classification has been made according to *K*_1_, in *P*_1_ and the area limited by the arc (*P*_1_*P*_3_*P*_4_*P*_2_) and the segments [*P*_2_*P*_8_] [*P*_3_*P*_1_], Area2, giving a clue on the size of the bottom part of the patient’s belly.
$$II_{1}\{\begin{array}{l} K_{1}\in \left[0.000194215;\;0.005002445\right]\\ A_{2}\in \left[7.56;94.89\right] \end{array}\;\mathrm{flat}\;\mathrm{profile}\;(\mathrm{FTP})$$The thoracic—abdominal profile curve is characterized by the fact that arcs (*P*_7_*P*_1_) and (*P*_1_*P*_3_), respectively are very far away from the vicinity of point *P*_1_, meaning that arc (*P*_7_*P*_3_) is almost the same with the segment [*P*_7_*P*_3_], that is the belly is straight at the front;
$$II_{2}\{\begin{array}{l} K_{1}\in \left[0.006819941;\;0.014204314\right]\\ A_{2}\in \left[91.35;187.54\right] \end{array}\;\mathrm{prominent}\;\mathrm{profile}\;(\mathrm{PP})$$The thoracic-abdominal profile curve is characterized by the fact that arcs (*P*_7_*P*_1_) and (*P*_1_*P*_3_), respectively are closer to the vicinity of point *P*_1_, meaning the belly is “pursed” towards the front part.IIThe third classification was done according to parameters $x_{4}-x_{2}$ (Anterior advancement) and $y_{4}-y_{2}$ (Inferior advancement)
$$III_{1}\{\begin{array}{l} Ant\mathrm{Pr}\in \left[22.39;69.16\right]\\ Ant\mathrm{Pr}\in \left[69.16;115.9\right] \end{array}\;\mathrm{anterior}\;\mathrm{projected}\;\mathrm{profile}\;(\mathrm{advanced})\;(\mathrm{AntPr})\;\mathrm{below}\;\mathrm{average}\;\mathrm{in}\;\mathrm{the}\;\mathrm{first}\;\mathrm{case}\;\mathrm{and}\;\mathrm{over}\;\mathrm{average}\;\mathrm{in}\;\mathrm{the}\;\sec\mathrm{ond};$$
$$III_{2}\{\begin{array}{l} Inf\mathrm{Pr}\in \left[2.41;42.85\right]\\ Inf\mathrm{Pr}\in \left[42.85;108.88\right] \end{array}\;\mathrm{inferior}\;\mathrm{projected}\;\mathrm{profile}\;(\mathrm{advanced})\;(\mathrm{InfPr})\;\mathrm{below}\;\mathrm{average}\;\mathrm{in}\;\mathrm{the}\;\mathrm{first}\;\mathrm{case}\;\mathrm{and}\;\mathrm{over}\;\mathrm{average}\;\mathrm{in}\;\mathrm{the}\;\sec\mathrm{ond}.$$

### 2.4. Data Processing and Statistical Analysis

#### 2.4.1. Data Processing

100 patients were diagnosed by ultrasound with hepatic steatosis at the Imagistic ward of the emergency clinical hospital Brasov. Males diagnosed by ultrasound with non-alcoholic fatty liver disease were considered eligible for the study if alcohol consumption was below 20 units/week while for the female subjects the consumption was limited at 14 units/week [28], in the absence of markers for infection with hepatitis B and C viruses and consumption of hepatic toxic medicine. 

Calculation of the liver diameters was performed based on the ultrasound technique recommended by World Health Organization WHO [29]. Each patient was measured the maximum anteroposterior diameters of right hepatic lobe (RHD) from the dome to the tip of the right lobe on the mediocavicular line and the maximum anteroposterior diameters of left hepatic lobe (LHD) on midline. The mathematical sum of the two diameters was called sum of hepatic lobes diameters (SHD). There also were measured: height, weight, body mass index (BMI) and waist circumference (WC).

Description of measured parameters is synthesized in Table 1.

There are some medical reasons to characterize only certain parameters of the thoracic-abdominal curve. Superior arc length and Osculating circle radius up are the parameters that characterize the chest, not the abdomen, and contribute to calculating Total area and Total arc length. Heigt and weight are both included in Body mass index.

The profile shape of the thoracic-abdominal curve has two sections with entirely different implications in evaluating abdominal adiposity: superior arc (SupArc) determined especially by the shape of the thorax with no contribution to abdominal adiposity and the inferior arc (InfArc) which is directly linked to the amount of fat on the abdomen. Similarly it can be assumed that Area2 has a substantial contribution in assessing abdominal fat by comparison with Area1, which depends mostly upon the thorax size. The parameters of interest that are analyzed in correlation with the amount of abdominal adiposity are checked in Table 1.

Because from the point of view of daily clinical practice it is difficult to reach a mathematical assessment of these curves, we tried creating 6 models of abdominal profiles that are easy to recognize during a simple inspection of the patient. These are: Raised profile (RP), Fallen profile (FP), Flat profile (FTP) and Prominent profile (PP), Anterior advanced profile (AntPr) and Inferior advanced profile (InfPr) (Figure 4).

Patients with “A type steatosic abdomen” were defined to present Prominent profile associated with Anterior advanced profile above average (PP + AntPr above average) and were compared to the patients with “A type non-steatosic abdomen” who presented Flat profile associated with Anterior advanced profile below average (FTP + AntPr below average).

Similarly, the patients with “B type steatosic abdomen” were defined by the occurrence of a Fallen profile associated with Inferior advanced profile above average (FP + InfPr above average) and they were compared to the patients with “B type non-steatosic abdomen” who presented Raised profile associated with Inferior advanced profile below average (RP + InfPr below average) (Figure 5).

Finally the volume resulted by rotating InfArc as a more accurate parameter of the real amount of fat. This was divided by 2 because in reality the patient’s belly develops only in an anterior direction. 

#### 2.4.2. Statistical Analysis of the Obtained Data

The normality assumption of the data was confirmed by help of the KS and SW tests. Data dispersion was evaluated by calculating the standard error of the mean (ErSt) or confidence intervals of 95% (95% CI) with “one sample *t*-test”. An independent-samples t-test was conducted to compare the difference between the SHD means. It was assumed there was equal variance of data when the Levene test was >0.05.

Parametric correlations were used (Pearson correlations, *r*) between anthropometric measurements and ultrasound measurements and the determination coefficient *R*^2^ for *r* was calculated. Where the existence of interference variables was assumed (weight, height, sex) between the correlated factors, partial correlations were analyzed after excluding these variables. The statistic significant correlations were analyzed by simple linear regression for the prediction of steatosis values by anthropometric parameters. A sample size of 100 patients achieves 90% power to detect a correlation coefficient of at least 0.316 (under the null hypothesis correlation of 0), using a two-sided hypothesis test with a significance level of 0.05. The statistic significance threshold was chosen as *p* < 0.05. The used software was SPSS 20.1, Excel 2013, G*Power 3.1, Adobe Illustrator CS6, JET_AreaLabel.jsx.

## 3. Results

### Results and Discussions

The studied group consisting of 100 patients included 50 men and 50 women, with the average age of 58.1 years (95% CI = 56.1; 60.0). No differences related to age were noticed in parameters analysis.

Anthropometric evaluations data stated that the mean height of the patients was 165.4 cm (95% CI = 163.3; 167.5), mean weight 89.8kg (95% CI = 86.5; 93.2), BMI average 32.9 (95% CI = 31.8; 34.0). In the studied lot there were statistically significant differences between men and women for height and weight (*p* < 0.001), but not for BMI (*p* = 0.068)

The correlations of interest between the geometric and ultrasound parameters are given in Table 2, where positive correlations are noticed, average in intensity and statistic significant, especially for SHD (directly linked to hepatomegaly) and InfArc, Area2, Vol. InfArc

Because the liver is bigger for men than for women and also for taller persons than for shorter persons [23], the influence determined by sex, weight and height upon SHD, as well as the geometric parameters of interest by assessing some “partial correlations” (control variables: sex, height and weight) (Table 3). Though the statistical power drops for Vol. InfArc, correlations remain valid for Area2 and InfArc.

Assessment by simple linear regression analysis revealed that InfArc of the thoracic-abdominal profile is the best predictor of hepatomegaly evaluated by SHD. (coefficient B = 0.17 (95% CI = 0.07; 0.26) *p* < 0.001) (Figure 6). Thus, increasing by 17 mm of the hepatic lobes diameters being associated with 100 conventional units growth of the inferior arc of the thoracic-abdominal profile. 

According to Table 4, it is possible to analyze the types of abdominal profiles. There are significant differences (*p* = 0.028) between the mean SHD for the patients with raised profile (mean SHD = 222.5 ± 4.5 mm) by comparison to the patients that do not present this profile (mean SHD = 236.1 ± 3.5 mm). In other words, patients with RP show smaller SHD distance with 13.6 ± 5.75 mm than the rest of the patients. Also, there are significant differences (*p* = 0.024) between mean SHD for the patients with prominent profile (mean SHD = 247.3 ± 7.2 mm) by comparison to the patients that do not share this type of profile (mean SHD = 229.2 ± 3.0 mm). In other words, patients with PP present higher SHD with 18.1 ± 7.9 mm than the rest of the patients (Table 4).

Analysis of patients with “A or B steatosis type abdomen” by comparison to the patient with “A or B non-steatosis type abdomen” as far as the hepatic lobes diameters means are concerned is synthesized in Table 5. For B category patients, there is a statistically significant difference (*p* = 0.025) regarding LHD size with 20.6 ± 8.7 mm in favor of steatosis profile (mean LHD = 172 ± 7 mm for steatosis profile vs. 151.4 ± 4.6 mm for non-steatosis profile). For B category patients there is a 16.4 ± 8.9 mm difference in favor of steatosis profile (mean SHD = 237.8 ± 8.3 mm for steatosis profile vs. 221.3 ± 4.8 mm for non-steatosis profile), but the statistically significant threshold is not to be reached (*p* = 0.071). 

## 4. Conclusions

The investigation method is an original one, without additional costs. In order to use this method, only a digital camera, a tripod and image acquisition software (Adobe Illustrator was used in the present paper) is required, followed by the overlap of thoracic-abdominal profile over Bezier’s model. 

The initial hypothesis that BMI-Body Mass Index is correlated with NAFLD [30] is confirmed, there is a positive correlation between the abdominal adiposity and the fat load of the steatosic fatty liver (as a form of visceral adiposity) and this correlation can be quantified by measuring some exterior parameters deducted from the calculus of thoracic-abdominal curve, like the inferior arc, Area2 and the volume of the inferior abdomen.

The size of the inferior arc can be used to deduce the size of the steatosic liver expressed by the sum of hepatic lobes diameters determined by ultrasound.

Certain types of abdominal conformations that are more probably associated with more important fat load of the steatosic liver can be defined (B type steatosic profile with the right hepatic lobe and A type steatosic profile with the sum of hepatic lobes diameters). These classifications can be used in diagnosing NAFLD stage; they are totally non-invasive and inexpensive methods. The method is original and can be implemented in any hospital, form any developed or less developed society, without high financial requirements.

We propose the design of a computerized software to measure the arc lengths resulted by overlapping the image acquired by digital camera and automated generation of the thoracic-abdominal profile in one of the I–IV classes (that is one of the above mentioned four types of abdomens: steatosic (type A or B), non-steatosic (type A or B).

The limits of the study are the relative small number of analyzed patients and the lack of a fitting procedure which optimizes the intermediate control points in order to minimize the least squared error, meaning the lack of an algorithm optimization procedure. We aim to develop a mobile application to quickly determine the profile of the belly and fit into the specified classification for the rapid diagnosis.

## Figures and Tables

**Figure 1 jcm-08-00065-f001:**
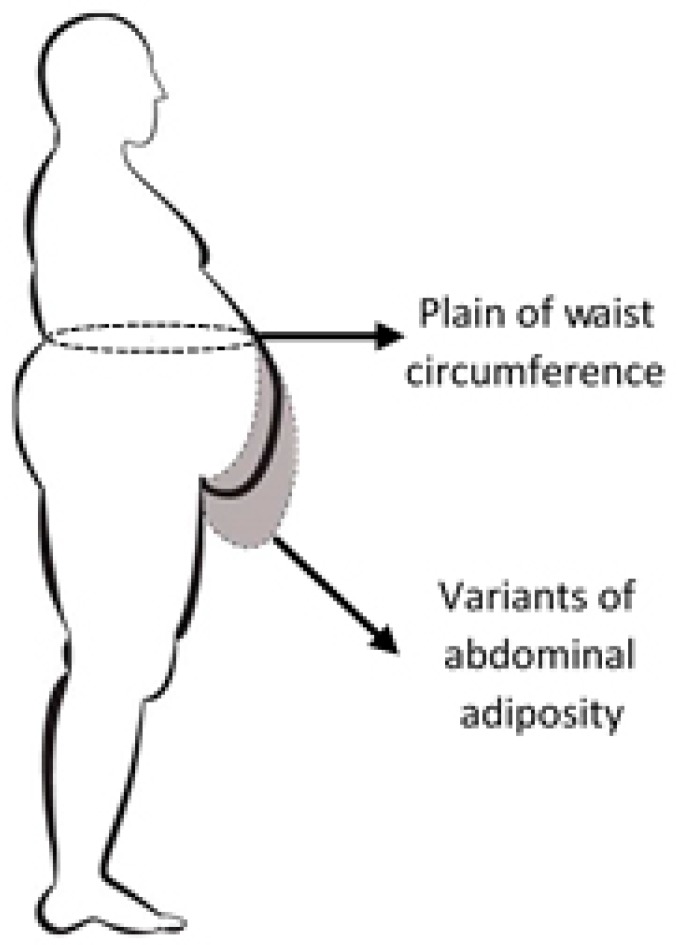
Thoracic-abdominal profile in case of non-alcoholic fatty liver disease (NAFLD) diagnosis.

**Figure 2 jcm-08-00065-f002:**
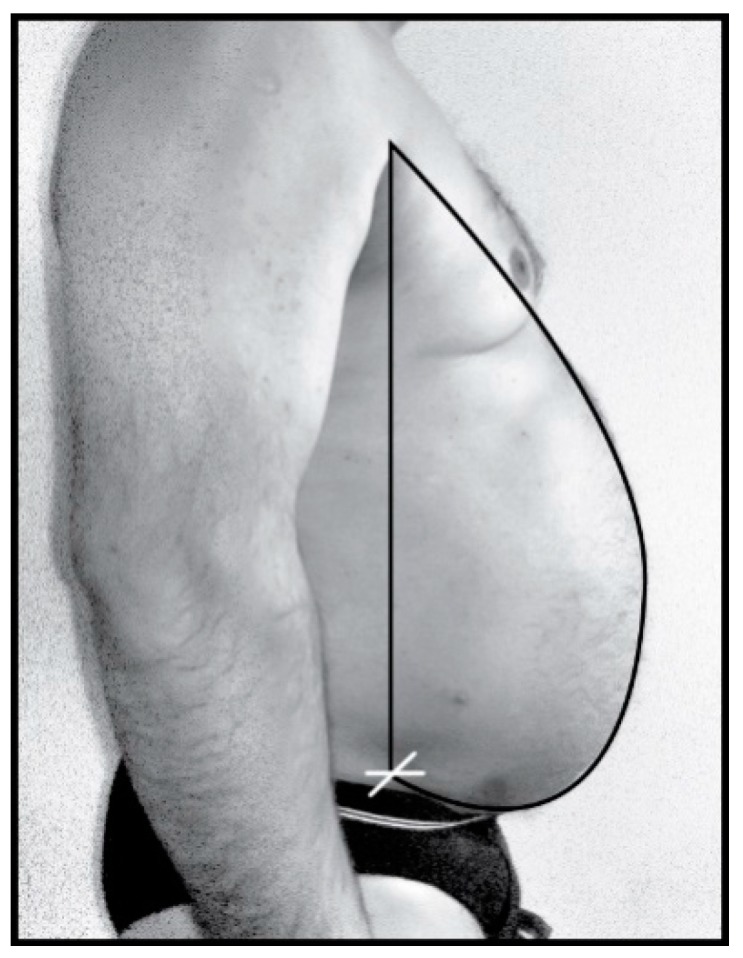
Thoracic-abdominal profile image acquisition.

**Figure 3 jcm-08-00065-f003:**
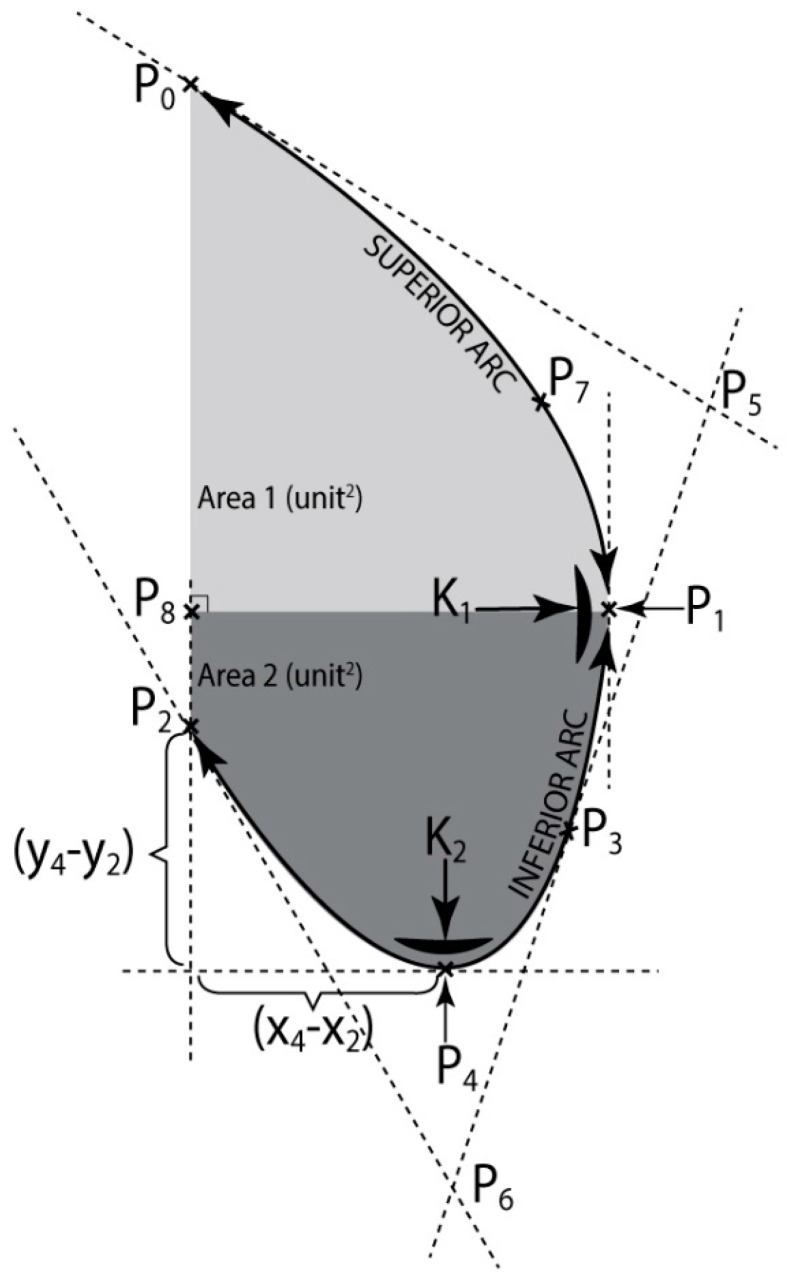
Mathematical modeling of thoracic-abdominal profile.

**Figure 4 jcm-08-00065-f004:**
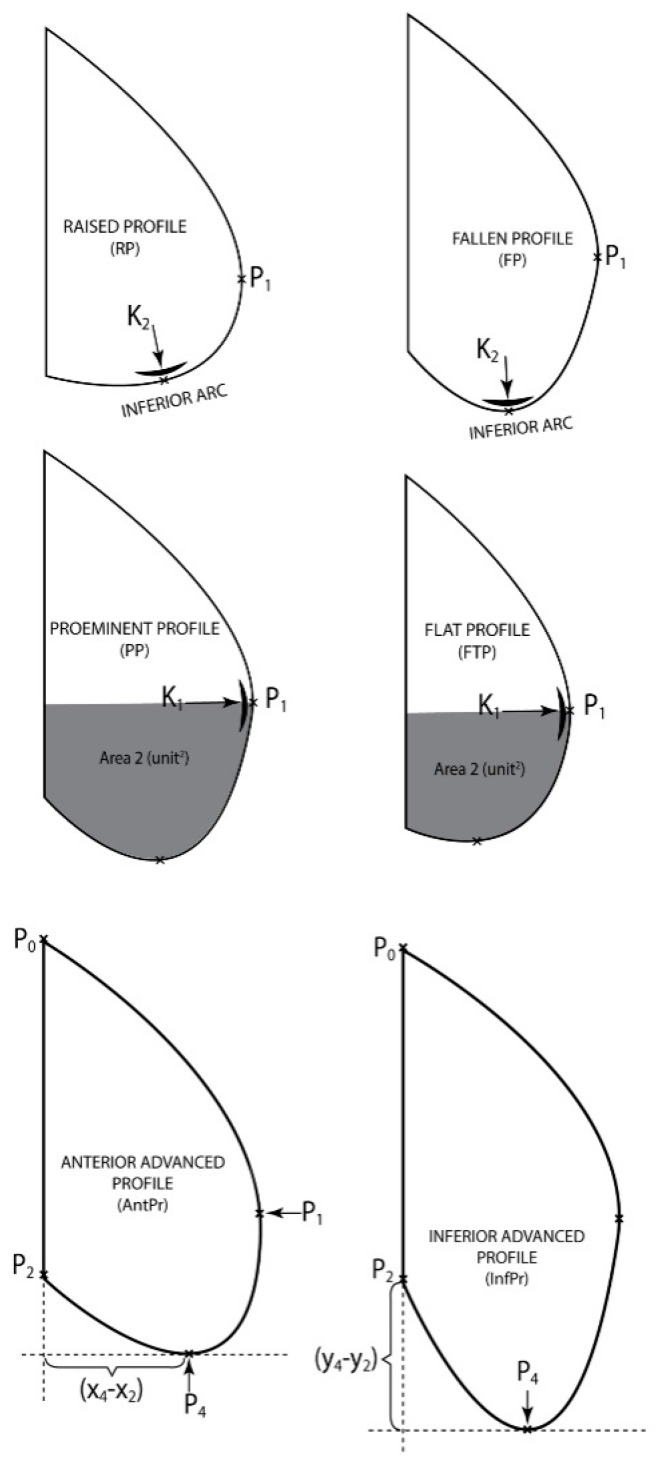
Models of abdominal profiles A.

**Figure 5 jcm-08-00065-f005:**
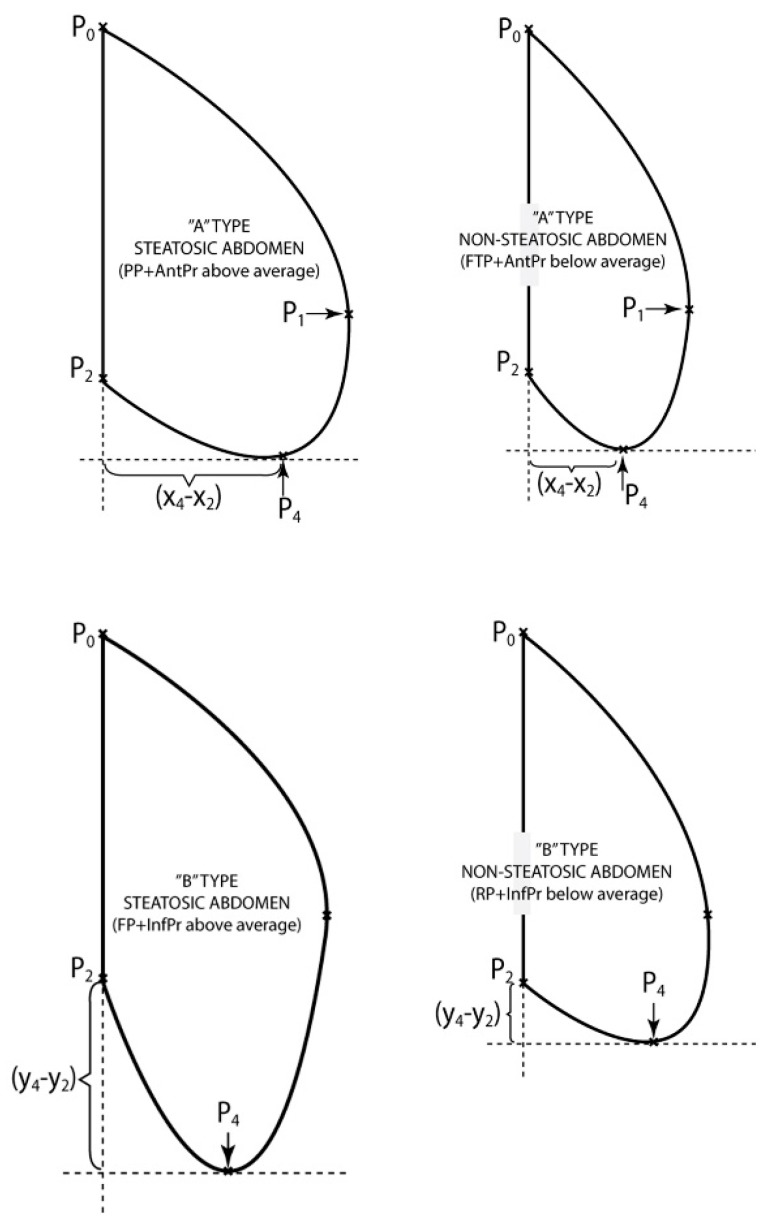
Models of abdominal profiles B.

**Figure 6 jcm-08-00065-f006:**
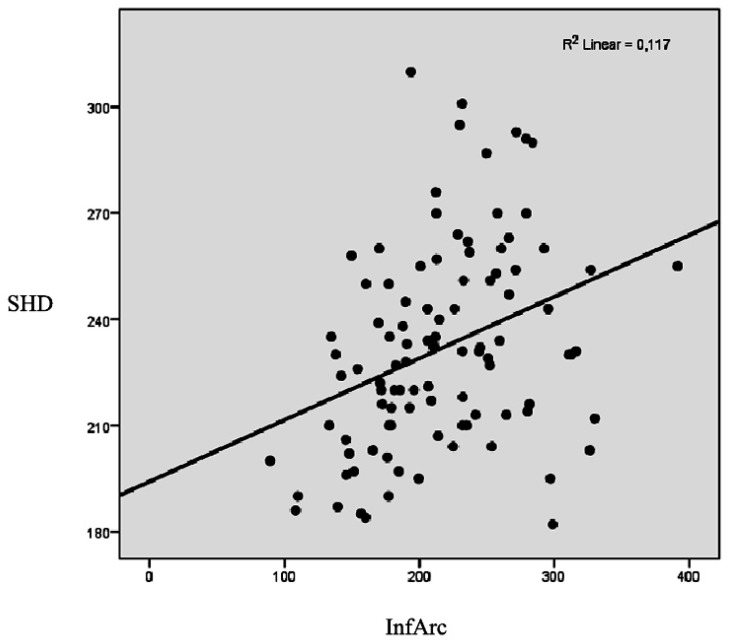
Linear regression between sum of hepatic lobes diameters (SHD) and InfArc parameters.

**Table 1 jcm-08-00065-t001:** The most important measured parameters.

Group Measurements	Measurements	Abbreviations	Parameters of Interest
Geometric parameters	Superior arc length	SupArc	
	Inferior arc length	InfArc	✓
	Total arc length	TotalArc	
	Superior curvature	K2	✓
	Osculating circle radius up	R.osc.up	
	Inferior curvature	K1	✓
	Osculating circle radius down	R.osc.down	
	Anterior push up	$x_{4}-x_{2}$	✓
	Inferior push up	$y_{4}-y_{2}$	✓
	Total area		
	Area above waist circumference	Area1	
	Area below waist circumference	Area2	✓
	Volume of area 2	Vol. InfArc	✓
Anthropological parameters	Height		
	Weight		
	Body mass index	BMI	✓
	Waist circumference	WC	✓
	Profile type (raised, fallen, flat, prominent, anterior advanced and inferior advanced)	RP, FP, FTP, PP, AntPr, InfPr	✓
Ultrasound parameters	Anterior diameter of the right hepatic lobe	RHD	✓
	Anterior diameter of the left hepatic lobe	LHD	✓
	Sum of hepatic lobes diameters	SHD	✓

**Table 2 jcm-08-00065-t002:** Statistical correlations between liver diameters and most significant measurements.

Correlations				
		RHD	LHD	SHD
WC	Pearson correlation (*r*)	0,.253 *	0.369 **	0.355 **
	Significance (*p*)	0.011	<0.001	<0.001
BMI	Pearson correlation (*r*)	0.159	0.197 *	0.209 *
	Significance (*p*)	0.113	0.049	0.037
K2	Pearson correlation (*r*)	0.087	−0.115	0.020
	Significance (*p*)	0.390	0.253	0.845
InfArc	Pearson correlation (*r*)	0.286 **	0.276 **	0.342 **
	Significance (*p*)	0.004	0.005	<0.001
Area2	Pearson correlation (*r*)	0.267 **	0.303 **	0.338 **
	Significance (*p*)	0.007	0.002	0.001
Vol. InfArc	Pearson correlation (*r*)	0.215 *	0.195	0.252 *
	Significance (*p*)	0.031	0.051	0.011

* Correlation is significant at the 0.05 level (2-tailed). ** Correlation is significant at the 0.01 level (2-tailed). WC: waist circumference, BMI: body mass index, K2: superior curvature, InfArc: inferior arc length, Area2: area below waist circumference, Vol. InfArc: volume of area 2, RHD: right hepatic lobe diameter, LHD: left hepatic lobe diameter, SHD: sum of hepatic lobes diameters.

**Table 3 jcm-08-00065-t003:** Partial correlations between SHD and most significant measurements, controlling the variables sex, height and weight.

Partial Correlations			
			SHD
Control Variables (Sex, Height, Weight)	InfArc	Correlation	0.283
		*p*-value	0.005
	Area2	Correlation	0.230
		*p*-value	0.024
	Vol. InfArc	Correlation	0.121
		*p*-value	0.239

**Table 4 jcm-08-00065-t004:** The mean difference of SHD between types of abdominal profiles.

		*t*-Test for Equality of Means	
		*p*-Value	Mean Difference
SHD	Raised profile (RP)	0.028	13.6
	Fallen profile (FP)	0.373	−6.7
	Flat profile (FTP)	0.237	7.1
	Prominent profile (PP)	0.024	−18.1

**Table 5 jcm-08-00065-t005:** The mean difference of liver diameters between types of steatosis profiles.

		*t*-Test for Equality of Means	
		*p*-Value	Mean Difference
Type A steatosis profile vs. Type A non-steatosis profile (PP + AntPr over average vs. FTP + AntPr below average)	RHD	0.119	−11.0
	LHD	0.167	−5.4
	SHD	0.071	−16.4
Type B steatosis profile vs. Type B non-steatosis profile (FP + InfPr over average vs. RP + InfPr below average)	RHD	0.025	−20.6
	LHD	0.905	0.7
	SHD	0.102	−20.0

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
