# Peer review of "Non-Invasive Assessment Method Using Thoracic-Abdominal Profile Image Acquisition and Mathematical Modeling with Bezier Curves"

_jcm, 2019, doi:10.3390/jcm8010065_

Reviewer 1 Report

Line 39. Do the authors mean Karlas et al.? Line 41. Define what TE fibrosis assessment is. Line 45. This sentence should not be a stand alone paragraph and needs a reference 1.1.1 is a good introduction, however the latter part of this section needs some re-writing and work to make it flow better to the reader and better define how NAFLD is currently diagnosed. Some further references are needed in this section. Line 59.  Does not need indenting. Line 61. Does not need to be a new paragraph. If so, please define “it”. Line 69. Introduces should be introduced Line 69. Occurred should be ?that occurred. Line 77. The sentence beginning Bioexponential does not make sense sorry. Line 78. Replace [14] with the names of the authors i.e. smith et al Section 1.2.2 needs referencing. Line 102. Fats deposits – should be ? fat deposits The first paragraph of 2.1 needs referencing. The paragraph beginning line 107 also needs referencing and is a little confusing to read, this could do with some re-wording. The sentence starting line 111 is too long, and needs to be made into more than one sentence. The paragraph beginning 120 needs referencing. 2.2. More information is needed here on how you identified the middle of the right iliac crest from a photograph. Line150. Should The be their? Line 157. Is should be are Section 2.3. Can you try to link these sentences rather than having multiple paragraphs of one sentence each. Line 247. The sentence beginning each patient needs re-wording. What do you mean by anterior diameter? Anterior to posterior or across the anterior of the liver from side to side. The process of the ultrasound needs more explanation. Line 253 needs explanation. Line 255. Entire should be entirely? Line 258 should read the amount of fat on the abdomen. Line 285 and 287 “t” Line 285. The sentence beginning The difference, - does not make sense grammatically . Line 306 statistic, should say statistically Line 312 – This sentence needs a reference. This is a good and well worthwhile endeavour and I commend the authors in thinking outside the square for a non invasive and cheap way to analyse NAFLD. There are some grammatical errors that needs addressing and some areas that require references. Some re-formatting of single sentence paragraphs into flowing bodies of text would enhance the paper. Can you justify why you measured the liver with ultrasound the way you did. Was this based on a previous study? The major flaw of this paper is the justification of the liver measurements and how/why they are considered hepatomegaly. Where did this protocol come from and how are you defining hepatomegaly from the ultrasound.

Author Response

Response to Reviewer 1 Comments

Point 1 : Line 39. Do the authors mean Karlas et al.?

Response 1: Yes, Karlas T, Petroff D, Garnov N, Bo¨hm S, Tenckhoff H, et al. (2014) Non-Invasive Assessment of Hepatic Steatosis in Patients with NAFLD Using Controlled Attenuation Parameter and 1H-MR Spectroscopy. PLoS ONE 9(3): e91987. doi:10.1371/journal.pone.0091987.

In Karlas et al study a head-to-head comparison performed between both methods is presented. In conclusion data suggest a comparable diagnostic value of CAP and 1H-MRS for hepatic steatosis quantification. Combined with the simultaneous transient elastography (TE) for fibrosis assessment, CAP represents an efficient method for non-invasive characterization of NAFLD. Limited correlation between CAP and 1H-MRS may be explained by different technical aspects, anthropometry, and presence of advanced liver fibrosis. [4]

Point 2: Line 41. Define what TE fibrosis assessment is.

Response 2: Combined with the simultaneous transient elastography (TE) for fibrosis assessment, CAP represents an efficient method for non-invasive characterization of NAFLD. Limited correlation between CAP and 1H-MRS may be explained by different technical aspects, anthropometry, and presence of advanced liver fibrosis. [4]

Point 3: Line 45. This sentence should not be a stand alone paragraph and needs a reference 1.1.1 is a good introduction, however the latter part of this section needs some re-writing and work to make it flow better to the reader and better define how NAFLD is currently diagnosed. Some further references are needed in this section.

Response 3: Limited correlation between CAP and 1H-MRS may be explained by different technical aspects, anthropometry, and presence of advanced liver fibrosis. [4]

Other radiological investigations for non-invasive diagnosis of NAFLD have been proposed, including computed tomography, magnetic resonance imaging and proton magnetic resonance spectroscopy [5], the controlled attenuation parameter using transient elastography [6] and Xenon 133 scan.[7]

Ultrasound is a non-invasive, widely available, and accurate tool in the detection of NAFLD[8] when sonographic features unique to NAFLD are standardized[9] and used to aid in diagnosis.

Point 4: Line 59. Does not need indenting.

Line 61. Does not need to be a new paragraph. If so, please define “it”.

Point 5: Line 69. Introduces should be introduced Line 69. Occurred should be ?that occurred.

Response 5

Waist circumference (WC) is a simple and inexpensive tool for assessing body fat distribution.

WC correlates well with abdominal obesity and it is associated with increased risk for adiposity-related morbidity and mortality.[4] WC and waist/hip ratio (WHR) are used as markers of abdominal obesity as they reflect central obesity.[15] It has been proposed that WHR and/or WC are more related to NAFLD than BMI. [16]

In summary, WHR and WC are simple tools that can be applied as important anthropometric indicators to screen populations with a high risk for NAFLD. [17]

Point 6:  Line 77. The sentence beginning Bioexponential does not make sense sorry.

Response 6 :  Bézier curves, bioexponential and cubic fitted to the Opto-Electronic

Point 7: Line 78. Replace [14] with the names of the authors i.e. smith et al Section 1.2.2 needs referencing.

Response 7:

The fitting procedure is described by Murthy et al. [20] and provides a set of smooth Bézier curves that describe the isopotentials. An alternative approach to curvature estimation involves precomputing the generic form of the curvature function for a Bézier curve. [20-21]

Point 8 :Line 102. Fats deposits – should be ? fat deposits The first paragraph of 2.1 needs referencing.

Response 8:

Bezier curves, mathematical formulation [21-22-23]

Let us consider                                               , n+ 1 distinct points in the Euclidean two-dimensional space E2, named control points or just controls. The polygon obtained by joining the control points starting with P0 and finishing with Pn is called control polygon or Bézier polygon. The control polygon is not unique.

Point 9: The paragraph beginning line 107 also needs referencing and is a little confusing to read, this could do with some re-wording.

Response 9: Properties of Bézier curves

1. The ends of the curve are tangent to the segments  respectively;

2. A Bézier curve can be divided into two Bézier curves in any of its points;

3. Bézier curve is entirely included in the convex cover of its control points;

4. A quadratic Bézier curve is a parabola segment;

5. By translating or rotating a Bézier curve, another Bézier curve is obtained.

Point 10: The sentence starting line 111 is too long, and needs to be made into more than one sentence.

Response 10:

The present study was conducted in accordance with the guidelines of Transilvania University Ethical Commission (Approval B21-2013) and under Spitalul Municipal Brasov, Romania.[25]

The paragraph beginning 120 needs referencing. 2.2. More information is needed here on how you identified the middle of the right iliac crest from a photograph. Line150. Should The be their?

Response 11: Direct evaluation of the amount of visceral and abdominal adiposity requires consuming and expensive imaging investigations (computed tomography, nuclear magnetic resonance etc.). This is the reason for searching indirect possibilities of measuring them. Currently, the most used parameter correlated with the amount of abdominal adiposity is waist circumference. WC measured according to WHO recommendations [30] in a plane which is parallel to the floor plane, at half the distance between the last rib and the iliac crest, at the end of expiration. The patient with empty stomach (à jeun), stays with close heels and relaxed abdominal muscles. This way to measure WC underestimates the relationship with abdominal obesity, the most part of adiposity being placed under the waist circle, due to different shapes of the “abdominal profile”. (Fig. 1)

Point 12: Line 157. Is should be are Section 2.3. Can you try to link these sentences rather than having multiple paragraphs of one sentence each.

Response 12: The mathematical model used in the present paper is based on the linked quadratic Bézier parametric curves, named paths in the graphics program, Adobe Illustrator.

Construction of a Bézier curve can be accomplished in two manners: either using geometric methods (Paul de Casteljau algorithm), or using algebraic methods (Bernstein form). The present paper uses the first recursive manner, by means of which starting form three points called control points, we aim at building curves that are as similar as possible to the curves resulted from the practical applications.

A limit of the considered mathematical model is the lack of a fitting procedure which optimizes the intermediate control points to minimize the least squared error, meaning the lack of an algorithm optimization procedure.

In parametric form each coordinate of a point on a curve is represented as a function of a single parameter [20]. Relation (3) is equivalent to the equations (4), meaning the parametric equations of a Bézier quadratic curve:

 Point 13:  Line 247. The sentence beginning each patient needs re-wording. What do you mean by anterior diameter? Anterior to posterior or across the anterior of the liver from side to side. The process of the ultrasound needs more explanation.

Response 13:

The thoracic-abdominal profile curve follows the arcs (P3P4), respectively (P4P2) very close in the vicinity of point P4 meaning the belly is very “pursed” at the bottom.

Point 14: Line 253 needs explanation.

Response 14:

The thoracic – abdominal profile curve is characterized by the fact that arcs (P7P1), respectively (P1P3) are very far away from the vicinity of point P1, meaning that arc (P7P3) is almost the same with the segment [P7P3], that is the belly is straight at the front;

Point 15: Line 255. Entire should be entirely? Line 258 should read the amount of fat on the abdomen.

Response 15:

The thoracic-abdominal profile curve is characterized by the fact that arcs (P7P1), respectively (P1P3) are closer to the vicinity of point P1, meaning the belly is “pursed” towards the front part.

I.           The third classification was done according to parameters x4-x2 (Anterior advancement) and y4-y2 (Inferior advancement)

Line 285 and 287 “t”

Response 16

Point 17: Line 285. The sentence beginning The difference, - does not make sense grammatically .

Response 17:

There are some medical reasons to characterize only certain parameters of the thoracic-abdominal curve. Superior arc length and Osculating circle radius up are the parameters that characterize the chest, not the abdomen, and contribute to calculating Total area and Total arc length. Heigh and Weigh are both included in Body mass index.

Line 306 statistic, should say statistically

Response 18:

In the studied lot there were statistically significant differences between men and women for height and weight (p < 0.001), but not for BMI (p = 0.068)

 Line 312 – This sentence needs a reference.

Response 19:

Because the liver is bigger for men than for women and also for taller persons than for shorter persons[24], the influence determined by sex, weight and height upon SHD, as well as the geometric parameters of interest by assessing some “partial correlations” (control variables: sex, height and weight). (Table 3) Though the statistical power drops for Vol. InfArc, correlations remain valid for Area2 and InfArc.

Reviewer 2 Report

This paper proposes the design of a computerized software to measure the arc lengths resulted by overlapping the image acquired by digital camera and automated generation of the thoracic abdominal profile in one of the I-IV classes. The topic is interesting and the paper is a piece of solid contribution in the field. However, the presentation can be improved. There are some technical issues that need to be addressed before I can give my final recommendation. The detailed comments are as follows.

(1) In Abstract, what are "disadvantaged" countries? I guess it should be developing countries. Am I right?
(2) In line 83, what is the definition of space E^2?
(3) It should be mentioned in Section 1.2.2 that the spirit of construction of Eq. (1) has inspired the proof techniques in opinion dynamics in social networks. See the two seminal works: Deffuant model with general opinion distributions: first impression and critical confidence bound; Deffuant model of opinion formation in one-dimensional multiplex networks.
(4) The contribution should be better motivated in the introduction. The novelty needs to be highlighted. For a paper to be published in JCM, a certain novelty is essential.
(5) In line 129, it is mentioned that the optical parallax errors were minimized and they will be neglected. An explanation is appreciated.
(6) In Fig. 3, what if the minimum point is at P_2 (rather than P_4)? Can Bezier curves method still be used?
(7) Why are there only three terms P_0, P_1, P_2 in Eq. (4)?
(8) It took me several sittings to figure out Eq. (18) for K(t). An explanation is appreciated. I think it is difficult for most readers of JCM.
(9) In the experiment results, Pearson correlation is used. It should be noted that this important coefficient has been widely applied in network theory. See the pertinent works: Geometric assortative growth model for small-world networks; Localized recovery of complex networks against failure.
(10) The conclusion can be improved by adding some future directions and possible investigation. This would be very helpful for the readers interested in this field.

Author Response

Response to Reviewer 2 Comments

 In Abstract, what are "disadvantaged" countries? I guess it should be developing countries. Am I right?

Response 1:The method is original, can also be implemented in developing countries without important financial resources and is totally non-invasive and cheap.
(2) In line 83, what is the definition of space E^2?

Response 2:

2.2. Bezier curves, mathematical formulation [21-22-23]

Let us consider                                               , n+ 1 distinct points in the Euclidean two-dimensional space E2, named control points or just controls. The polygon obtained by joining the control points starting with P0 and finishing with Pn is called control polygon or Bézier polygon. The control polygon is not unique.
(3) It should be mentioned in Section 1.2.2 that the spirit of construction of Eq. (1) has inspired the proof techniques in opinion dynamics in social networks. See the two seminal works: Deffuant model with general opinion distributions: first impression and critical confidence bound; Deffuant model of opinion formation in one-dimensional multiplex networks.

Response 3:

Observation 1. The spirit of construction of relation (1) has inspired the proof techniques in opinion dynamics in social networks. [24]
(4) The contribution should be better motivated in the introduction. The novelty needs to be highlighted. For a paper to be published in JCM, a certain novelty is essential.

Response 4:

The method of diagnosis is based on an easily reproduced experiment, it is original, innovative, non-invasive, and cost-effective. Can be implemented anywhere in the world, there is no need for investment, only for determining the profile of the belly.
(5) In line 129, it is mentioned that the optical parallax errors were minimized and they will be neglected. An explanation is appreciated.

Response 5:

WC measured according to WHO recommendations [30] in a plane which is parallel to the floor plane, at half the distance between the last rib and the iliac crest, at the end of expiration. The patient with empty stomach (à jeun), stays with close heels and relaxed abdominal muscles. This way to measure WC underestimates the relationship with abdominal obesity, the most part of adiposity being placed under the waist circle, due to different shapes of the “abdominal profile”. (Fig. 1)
(6) In Fig. 3, what if the minimum point is at P_2 (rather than P_4)? Can Bezier curves method still be used?

Response 6:

Observation 3. If the minimum point is at P2 instead of P4, then Bezier curves method can still be used analogously.
(7) Why are there only three terms P_0, P_1, P_2 in Eq. (4)?

Response 7:

In parametric form each coordinate of a point on a curve is represented as a function of a single parameter [20]. Relation (3) is equivalent to the equations (4), meaning the parametric equations of a Bézier quadratic curve:
(8) It took me several sittings to figure out Eq. (18) for K(t). An explanation is appreciated. I think it is difficult for most readers of JCM.

Response 8:

where , i=1,2.
(9) In the experiment results, Pearson correlation is used. It should be noted that this important coefficient has been widely applied in network theory. See the pertinent works: Geometric assortative growth model for small-world networks; Localized recovery of complex networks against failure.

Response 9:

We completed with this work.
(10) The conclusion can be improved by adding some future directions and possible investigation. This would be very helpful for the readers interested in this field.

Response 10:

We aim to develop a mobile application to quickly determine the profile of the belly and fit into the specified classification for the rapid diagnosis.

Round  2

Reviewer 2 Report

I appreciate the authors' responsiveness and careful revision. The revised version is a pleasant reading.